# Benchmarking Long-Term Memory with Continuous Dialogue Lifelogs

## Abstract

Memory system in the real world holds considerable promise, especially in the potential continuous dialogue lifelogs scenarios, where wearable devices with microphone always-on can keep recording the surrounding dialogue. Existing benchmarks mostly focus on Person-AI interaction or Person-Person conversations, neglecting the continuous dialogue lifelogs scenarios, integrating multi-person interaction, causal and temporal event threads and so on. In this paper, we propose two benchmark, named **EgoMemBench** and **LifeMemBench**, with a hierarchical life simulation framework. EgoMemBench is built in a bottom-up manner from a real-world lifelogging video dataset EgoLife over a seven-day period, while LifeMemBench is simulated by LLMs with a top-down elaboration to generate year-long personal lifelogs. Based on the hierarchical data with different temporal granularities, we design an automatic question-answering construction pipeline to generate four types with high-quality. Regarding the evaluation mode, employing both online and offline approaches–with the online mode prioritized, as it better aligns with the continuous dialogue lifelogs scenario. Experiments across four representative memory systems show that MemOS consistently outperforms others, achieving overall accuracies of 67.59% and 66.16% on the benchmarks. This highlights the value of fine-grained memory management and the effectiveness of our benchmarks. Moreover, we show that event-level semantic segmentation of continuous dialogues yields superior results compared to naive chunking, pointing to more effective ways of structuring lifelog memories. In conclusion, we define a continuous dialogue lifelogs scenario, positioning it as a potential cornerstone for next-generation terminal AI assistants.

## 1 Introduction

Large language models (LLMs) have demonstrated remarkable capabilities across a wide range of tasks (OpenAI, 2022; OpenAI et al., 2024; Yang et al., 2025a), especially in the single-turn scenario with short-term conversational context. Subsequently, LLMs show superior reasoning ability as automatic agents to process a series of complex tasks in real world (Schick et al., 2023; Yang et al., 2023), meanwhile, place a higher requirements on the context length. To explore long-term memory capability of LLMs, one line of works (Chen et al., 2024; Grattafiori et al., 2024; Yang et al., 2025a) focus on probing the accuracy of locating evidence in extremely long-context passages, such as Needle In A Haystack (NAIH). However, the strategy of increasing context length indefinitely is not a solution to long-term memory, due to the exponential growth in inference costs and the ability of long-term memory utilization (Hsieh et al., 2024; Li et al., 2024; Liu et al., 2024). Consequently, the development of memory system has emerged. It requires LLMs to adaptively remember and retrieve relative evidence from massive information over extended periods.

Meanwhile, there exist various benchmarks primarily focus on dialogue scenarios, covering Person-AI interaction (Jiang et al., 2025; Wu et al., 2024) and dyadic dialogues (i.e., person-person conversations; Maharana et al. (2024)). However, the above-mentioned studies neglect a promising scenario as illustrated in Figure 1: *continuous dialogue lifelogs*. Nowadays, there emerges a series of commercial wearable devices with potential to achieve microphone always-on, such as smart glasses (e.g., Ray-Ban Meta, RayNeo V3/X3, Xiaomi AI Glasses), and recording machines (e.g., Plaud). Equipped with these wearable devices, users can continuously record the surrounding audio which fully filled with intensive dialogue content. Using automatic speech recognition (ASR), the

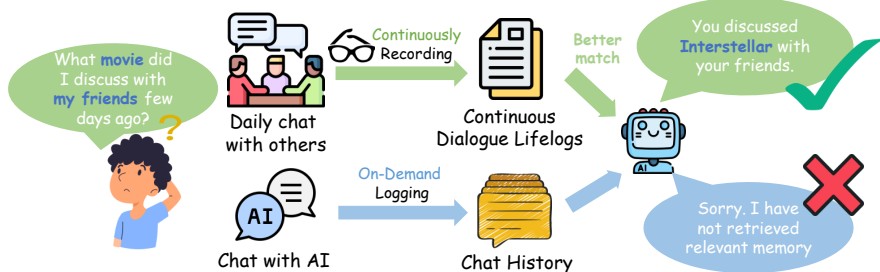

Figure 1: Comparison between (1) The microphone-always-on scenario, which continuously recording dialogue with others in daily life, and (2) Chatting with AI scenario, which on-demand logging to form the chat history.

audio stream is transcribed into text and stored in a long-memory database after post-processing. Compared to prior passages and Person-AI interaction, continuous dialogue lifelogs have several unique characteristics: (1) The daily conversations integrate multi-person interactions, casual and temporal event threads, and simulated social networks. (2) Through round-the-clock recording, the lifelogs enables the AI assistant to accumulate an extensive understanding of users' facts, perfectly embodying the highly promising usage scenario of an personalized assistant.

To systematically evaluate the long-term memory capacity of agents in continuous dialogue lifelogs, we introduce two complementary benchmarks as shown in Figure 2: **EgoMemBench** and **LifeMemBench**. Both benchmarks adopt a hierarchical life simulation framework to generate the dataset. EgoMemBench is constructed using a bottom-up (i.e., from second to week) summarization based on the real-life first-person video dataset EgoLife (Yang et al., 2025b), which records egocentric video from six individuals over a seven-day period. To extend the temporal span and ensure long-horizon coherence, we further use LLMs with a top-down (i.e., from year to day) elaboration to simulate a year-long personal lifelog rich in multi-party conversations, forming the LifeMem-Bench. For both benchmarks, we generate QA pairs from multi-level event summaries, enabling systematic probing of memory retrieval across different temporal granularities. Notably, we first propose a **online evaluation** protocol that follows the linear flow of time with information update and conflict, offering a more realistic assessment of long-term memory in real-world conditions.

In experimental results, we evaluate four representative memory systems based on `Qwen3-8B` on both EgoMemBench and LifeMemBench, yielding critical insights into lifelog memory system design. MemOS consistently outperforms all baselines, with its vector-based variant (MemOS-V) often matching or exceeding the graph-based (MemOS-G) design. This phenomenon challenges the assumption that complex structured storage is indispensable and underscores the value of fine-grained memory management. Notably, several state-of-the-art approaches underperform a simple RAG baseline, finding that reinforces the criticality of preserving raw textual evidence in lifelog scenarios. The most pronounced challenge across all methods emerged in temporal retrieval tasks, which require precise timestamp alignment-a core lifelog capability that remains underexplored in prior work. Under our proposed online evaluation protocol, where difficulty escalates gradually with extended interaction horizons, systems demonstrate improved performance. This reflects the realism of continuous lifelog dynamics. Finally, our results validate that event-level semantic segmentation of continuous dialogues significantly outperforms naive chunking strategies, offering a clear pathway for optimizing lifelog memory structuring. Collectively, these findings establish the dual importance of preserving raw context and implementing intelligent memory organization for next-generation lifelog-aware systems.

## 2 RELATED WORK

**Memory Systems.** The architecture of memory systems can be summarized as Figure 3. The system collaborates with a chat agent, usually containing a summary agent to summarize memories (Xu et al., 2025), a memory manager to manage the database (Chhikara et al., 2025), a retriever for searching, and a database stores memories. Several works have implemented this framework in various ways. Wang et al. (2025),Xu et al. (2025), and Chhikara et al. (2025) used a Summary Agent to condense memories before storing them in a vector database. While Chhikara et al. (2025),

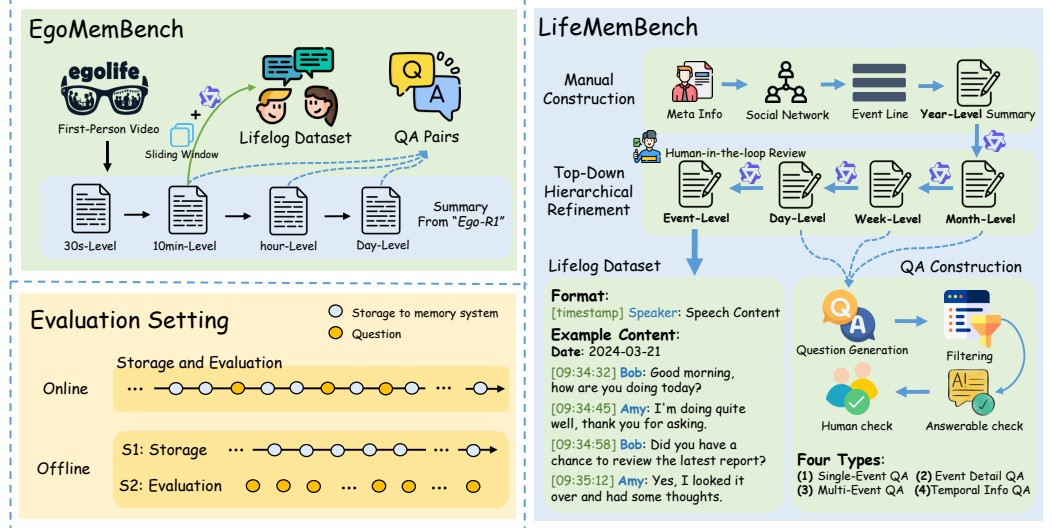

Figure 2: We propose two benchmarks: EgoMemBench (top right), constructed from real-world egocentric videos (EgoLife), and LifeMemBench (left), a more comprehensive benchmark built upon Top-Down Hierarchical Life Simulation Framework. We also introduce a novel online evaluation method that assesses performance incrementally during data storage, unlike conventional end-of-storage evaluation.

Gutiérrez et al. (2025), and Rasmussen et al. (2025) maintain a graph rather than vector. Other approaches borrow ideas from operating system to manage the memories, such as Packer et al. (2023) and Li et al. (2025).

**Benchmarks for Long-Term Memory.** With the development of memory systems, several benchmarks are developed for evaluation. These benchmarks aimed at different application scenarios, as shown in Table 1. Zhong et al. (2024) manually constructed 97 QA-pair for simple QA. LoCoMo (Maharana et al., 2024) creates a long-term dataset for Person-Person dialogue and evaluates the ability on different dimensions. Jiang et al. (2025) creates a larger dataset for Person-AI dialogue with a context length of 1M. LongMemEval (Wu et al., 2024) further expanded the data scale, with each dialogue record containing 500 sessions and up to 1.5M tokens. They made well efforts for chatbot-like memory system, however, remaining a gap between real world scenarios–the scenarios of multi-person communication, and the situation where the memory system is continuously activated, such as an always-on personal agent. In this paper, we purposed a benchmark contains multi-person dialogue, which is rolling from day to night, and continuous for year-long. This benchmark behaves closer to the real world compared to previous works.

## 3 BENCHMARKS

To explore the long-term memory capacity of agents in continuous dialogue lifelogs, we specifically construct two complementary benchmarks for egocentric memory (Cheng et al., 2024), as illustrated in Figure 2. The first benchmark, named **EgoMemBench**, is constructed based on the existing Ego-Life dataset (Yang et al., 2025b) which contains daily video recording across seven days. Moreover, to mimic the continuous dialogue lifelogs in real-life with more time span and scene diversity, we further adopt data synthesis to construct a year-long benchmark, named **LifeMemBench**. Both benchmarks are constructed with a hierarchical life simulation framework, where use bottom-up and top-down manners due to different data source. More details will be introduced in this section.

### 3.1 EGOMEMBENCH

**Data source.** We construct our dataset based on the Ego-R1 summarization corpus (Tian et al., 2025), which is derived from the EgoLife dataset (Yang et al., 2025b). EgoLife consists of over 300 hours of real-world, first-person recordings collected from six participants living together, each wearing Meta Aria smart glasses to capture approximately eight hours of egocentric video and audio per day for one week. Built upon this foundation, Ego-R1 organizes the raw data into multi-

Table 1: Comparison of memory benchmarks. The key properties are summarized including the type of scenario (**Scenario**), the temporal coverage (**Time Span**), number of sessions (**#Sessions**), whether continuous recording is supported (**Cont. Rec.**), whether the queries contain explicit timestamp for simulation (**TS**), whether support for online evaluation (**Online**).

| Benchmark | Scenario | Time Span | #Sessions | Cont. Rec. | TS | Online |
|---|---|---|---|---|---|---|
| LoCoMo (Maharana et al., 2024) | Person-Person | Few months | 1k | ✗ | ✗ | ✗ |
| MemoryBank (Zhong et al., 2024) | Person-AI | 10 days | 300 | ✗ | ✓ | ✗ |
| LongMemEval (Wu et al., 2024) | Person-AI | N/A | 50k | ✗ | ✓ | ✗ |
| MemBench (Tan et al., 2025) | Person-AI | N/A | 65k | ✗ | ✗ | ✗ |
| EgoMemBench | Multi-Person | 7 days | 1.7k | ✓ | ✓ | ✓ |
| LifeMemBench | Multi-Person | 1 year | 3.8k | ✓ | ✓ | ✓ |

scale textual summaries through a hierarchical pipeline: 30-second clips are first described by a Vision-Language Model (VLM), and these fine-grained descriptions are progressively aggregated into 10-minute, 1-hour, 1-day, and 1-week summaries. Rather than relying on direct transcription from EgoLife, we leverage the structured 10-minute summaries from Ego-R1 as prompts for a large language model (LLM) to generate plausible multi-turn dialogues. This generative strategy expands the dataset scale beyond the original recordings while preserving the semantic fidelity and temporal coherence of the egocentric narratives.

**Data Curation.** There are two critical challenges for data curation: (1) the inherent scarcity of long-form, naturally occurring lifelog data, and (2) the need to maintain narrative coherence and information density across extended temporal horizons. To this end, in the *granularity selection*, we choose the 10-minute summary level as a deliberate compromise. The reason is that finer-grained 30-second summaries are too fragmented to support coherent dialogue generation, while coarser summaries (i.e., 1-hour or 1-day) lead to superficial content and a loss of episodic detail. Crucially, we find that generating dialogues from concatenated 10-minute segments produces more information-rich outputs than direct generation from hour-long summaries, a phenomenon we attribute to the limited attention span of current LLMs (Mudarisov et al., 2025). The 10-minute granularity thus maximizes the utility of the data for downstream memory tasks. Then, we transform the 10-minute summary to our lifelog dataset. Specifically, we design a *sliding-window generation strategy* to ensure narrative continuity. Rather than directly generate lifelog dataset based on each separated 10-minute segment, we ask the LLMs to additionally conditioned on the six preceding segments, using the prompt as shown in Appendix D. This 60-minute context window is empirically determined to balance two needs: providing sufficient history to maintain speaker consistency and topic flow, while avoiding excessive repetition and staying within practical computational limits.

**Data Review.** We employ a hybrid quality assurance process combining human annotators and LLM assistance. Annotators evaluate dialogues for (1) naturalness, (2) coherence across segments, and (3) factual consistency with source summaries. Following the *text-grad*(Yuksekgonul et al., 2024) paradigm, the LLM first flags issues and provides targeted feedback, which are then used by annotators to revise the text, selectively accepting improvements. This iterative workflow, combining LLM feedback and human verification, provides an efficient review process that produces coherent, accurate, and high-quality data.

## 3.2 LIFEMEMBENCH

While EgoMemBench is constructed from real-life recordings, it has several limitations: (i) The dataset spans only seven days of daily activities, which is insufficient to capture long-term patterns of individual's daily life. (ii) It lacks the diversity of social contexts and location changes characteristic of real-world scenarios. To complement EgoMemBench and enable the study of long-term memory phenomena at scale, we therefore simulate continuous dialogue lifelogs of an individual, forming LifeMemBench. Our goal is to establish a more comprehensive and scalable benchmark that reflects longitudinal dynamics of daily life, incorporating realistic routines, diverse social interactions, and natural scene transitions over extended periods.

**Data Curation.** In order to simulate real *social networks*, we begin by constructing a virtual user profile and the corresponding social relationship. The user profile specifies demographic attributes (e.g., age, occupation), while the comprehensive social relationship network encompassing family,

| | | A-mem | MemO | MemOS |
|---|---|---|---|---|
| Summary | | ✓ | ✓ | ✓ |
| Manager | | save & update | save, update, delete | save, update, delete |
| Retriever | | direct query | query, graph search | query + OS |
| Chat Agent | | outer | outer | inner/outer |
| Database | | vector | vector, graph | text, graph, vector |

Figure 3: Definition of the structure of a memory system, and a comparison table of current memory system approaches under this structure.

colleagues, and friends. Instead of directly generate high-level annual experience based on this simulated social network, we first construct eleven *event lines* for the year across key life dimensions, such as work and family. These event lines serve as structured narrative threads that capture diverse and realistic life dynamics. By anchoring the simulation in carefully designed event structures, we establish a solid basis for generating lifelog data that maintains semantic richness and long-horizon coherence. We then align all event lines into *year-level summary*, and further employ *a top-down refinement strategy* to progressively generate lifelog narratives at monthly, weekly, and daily scales. Each refinement proceeds in two stages: (i) allocation, where we use LLMs to distribute high-level descriptions into lower-level placeholders (e.g., mapping annual events into monthly summaries); and (ii) enrichment, where LLMs expand each placeholder into detailed narratives constrained by the higher-level context. To further align the simulated lifelog with real-world temporal structures, we incorporate external calendar signals such as statutory holidays, weekends, and workdays. Through iterative allocation and enrichment, we obtain richly detailed *monthly*, *weekly*, and *daily experience* descriptions that preserve both narrative continuity and contextual realism. However, directly generating lifelogs based on daily experience often leads to coarse or repetitive descriptions, as LLMs cannot capture the fine-grained variations that naturally arise within a day. To address these issues, we segment daily descriptions into *event-level narratives*, each annotated with temporal boundaries, locations, and participants. Finally, we generate the *continuous dialogue lifelogs*. Each dialogue is conditioned on the event context, the virtual user's background, and the social relationship network, ensuring natural conversational flow. Applying this framework, we obtain a year-long lifelog comprising rich, dialogue-centric daily records. Beyond the dataset itself, this framework establishes a scalable methodology for simulating long-term, always-on scenarios, providing a foundation for benchmarking memory-intensive AI assistants in realistic yet privacy-preserving settings.

**Data Review.** During the top-down generation process, data from each stage undergo human inspection and revision before proceeding to the next stage. Only when the quality of the current stage passes the review does the data advance to the subsequent stage. For example, "monthly experience descriptions" must first pass manual quality checks before "weekly experience descriptions" are generated. This iterative verification reduces hallucinations and logical inconsistencies, while keeping the overall cost of human involvement manageable.

### 3.3 QUESTION-ANSWERING PAIRS CURATION

**Task Format.** To elicit the stability and usability of assessment, we officially opt for *multiple-choice* format over open-ended question-answering (QA). For each question, we provide one ground-truth answer and three distractor options: the question serves as a query, tasking the system with retrieving relevant memories to select the optimal choice. This setup evaluates the memory system's capacity to store, manage, and retrieve memories. While the benchmark can be transformed to support open-ended QA, with evaluation conducted via LLM-as-a-Judge (Zheng et al., 2023), which aligns more closely with real-world scenarios and further assesses the chat agent's ability to organize information and generate clear responses.

**Question Types.** The primary question types in the memory system (Maharana et al., 2024) can be categorized into three main classes: temporal reasoning, event recall, and detail retrieval. Temporal reasoning requires agents to integrate information from multiple event fragments to derive reasoned conclusions. Event recall involves ambiguous queries, where agents need to mine lifelog data to retrieve the most relevant contextual evidence. Detail retrieval focuses on pinpointing specific event attributes, often buried in dense lifelog streams, requiring precise snippet-level recall. Based

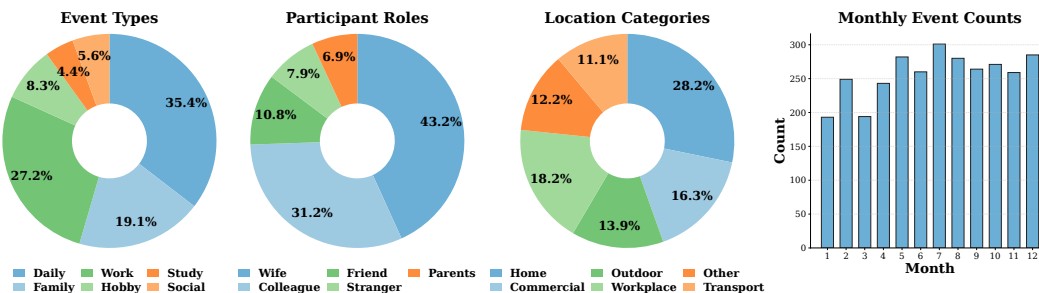

Figure 4: Distributional statistics of the LifeMemBench dataset. The plots summarize event types, social roles, locations, and monthly dialogue counts, showing that the dataset is balanced and closely aligned with real-world lifelog patterns.

on the categories above, we further refine our design to construct four distinct question types: (i) *QT1: Event Content Recall*: This type encompasses questions that demand the retrieval of core event content, and it falls under the broader category of Event Recall. (ii) *QT2: Event Detail Retrieval*: Questions of this type require precise retrieval of specific event details, and they are classified under Detail Retrieval. (iii) *QT3: Multi-hop Event Reasoning*: These questions involve both retrieving and reasoning across multiple events, and they belong to the Temporal Reasoning category. (iv) *QT4: Temporal Information Question Answering*: As a lifelog-specific subcategory of Detail Retrieval, this unique type requires accurately pinpointing the exact timestamp of a particular event to generate a valid answer.

**Question-Answering Construction.** Based on the constructed EgoMemBench and LifeMem-Bench with hierarchy structure, we propose an automatic pipeline to generate question-answering (QA) pairs as shown in Figure 2. We use 10-minute, hour-level and day-level in EgoMem-Bench to construct QA pairs, while use day-level, week-level, and month-level for LifeMem-Bench. As a result, the generated QA pairs can span over different temporal granularities. Based on these data, we prompt `Qwen3-235B-Thinking-2507`[1] (Yang et al., 2025a) to generate the expected types of QA pairs and the corresponding distractors. The query timestamp is set a few days after the evidence timestamp, and manually added into the question's metadata.

However, the generated QA pairs often contain information leakage (e.g., timestamps). This problem enables agents to "cheat" in answering, as they only need to retrieve based on the timestamp in the query rather than truly understanding the question. Therefore, we prompt a `Qwen3-32B` (Yang et al., 2025a) to filter out questions with potential information leakage, then erase such leakage and rebuild those questions. Moreover, considering that certain questions may be unanswerable, we further design an answerability check process. Specifically, we concatenate the lifelog, the question,

Table 2: The number of generated questions, the number of questions remained after filtering and human annotation, and the model accuracy in the final answerable verification.

| LifeMem | Total | Filted | Keep Rate | Model Acc. |
|---------|-------|--------|-----------|------------|
| Daily   | 1464  | 1430   | 97.68%    | 99.23%     |
| Weekly  | 248   | 241    | 97.18%    | 98.76%     |
| Monthly | 48    | 46     | 95.83%    | 100%       |
| All     | 1760  | 1717   | 97.56%    | 99.18%     |

and the corresponding options, then put this combined context into `qwen3-max-preview` to generate an answer. For questions answered incorrectly by the LLM, human annotators are tasked with screening them further: questions deemed unanswerable by annotators are marked and discarded, while those for which annotators can identify the correct option are labeled as answerable, retained, and categorized as "hard" questions. Table 2 reports the total number of questions generated in the construction stage, and the final questions in benchmark after applied LLM filtering and human annotation. The model accuracy shows that `qwen3-max-preview` is managed to answer nearly all questions correctly with evidences provided, indicating that the generated questions are answerable. After filtering, EgoMemBench obtains 1774 questions, while LifeMemBench obtains 1717 questions. The detailed prompts for question generation are discussed in Appendix D.1.

**Evaluation Modes.** Due to the unique character of continuous dialogue lifelogs scenario, we define two different evaluation modes, including offline and online. *Offline mode* is traditional evaluation mode in previous memory benchmarks (Maharana et al., 2024; Wu et al., 2024), where the memory

---

[1]https://huggingface.co/Qwen/Qwen3-235B-A22B-Thinking-2507

agent answers all questions at once after the memory system has processed all the data. Although widely adopted for evaluation, this paradigm differs fundamentally from real-world daily interactions, where user queries emerge spontaneously rather than being constrained to post-conversation intervals. To better align with continuous dialogue lifelogs, we propose a novel evaluation mode, termed *online mode*. As illustrated in Figure 2, the online mode operates across the temporal dimension in a streaming fashion: as time progresses, the memory agent answers queries using memories from the current timestamp, while the memory system concurrently processes incoming data. This design more closely emulates real-world memory systems, where users do not abruptly halt memory updates to pose all questions at once. Instead, they interleave queries about past events with continuous updates of new information to the memory system.

## 3.4 STATISTICS

Figure 4 illustrates the distributional characteristics of our synthetic dataset LifeMemBench, encompassing four key dimensions: event types, social roles, locations, and monthly dialogue counts. Overall, the dataset exhibits a balanced structure that aligns with realistic lifelog patterns. The event distribution spans both routine necessities and higher-level pursuits, while the social role distribution includes intimate interactions, professional contexts, and casual engagements. Geographically, conversations are distributed across diverse settings such as homes, workplaces, transportation hubs, and outdoor spaces, further grounding the dataset in real-world scenarios. Temporally, monthly event counts remain stable, with no significant seasonal bias. This stability is particularly valuable for long-horizon memory evaluation, as it avoids skewing results toward time-specific patterns. Collectively, these properties establish LifeMemBench as a reliable testbed for memory agents: its balanced coverage minimizes distributional skew, while its diversity (across events, roles, and locations) and temporal regularity enable robust benchmarking under the "always-on" conditions that characterize real-world lifelog interactions. For the QA pairs, Figure 7 in Appendix D.2 displays the specific porportion of each QA type (introduced in Section 3.3) that remained after the filtering process. In LifeMemBench, the four question types—event content recall, event detail retrieval, multi-hop event reasoning, and temporal information QA—are distributed as 25.3/25.0/25.0/24.6. In EgoMemBench, their proportions are 25.1/25.1/24.9/24.9. The balanced distribution of question types enables a more comprehensive evaluation of memory systems' multifaceted capabilities.

## 4 EXPERIMENTS

### 4.1 EXPERIMENTAL SETUP

We extensively evaluate EgoMemBench and LifeMemBench across a suite of representative memory systems, whose overall architecture is illustrated in Figure 3. Specifically, we select four memory systems for assessment: (1) **RAG** (Lewis et al., 2021): A straightforward retrieval-augmented generation (RAG) baseline that directly stores and retrieves text chunks; (2) **A-Mem** (Xu et al., 2025): An enhanced RAG variant that augments stored text with additional semantic signals; (3) **Mem0** (Chhikara et al., 2025): A paradigm that stores structured observations distilled from raw text, rather than verbatim conversational history; and (4) **MemOS** (Li et al., 2025): A framework that manages memories via a memory operating system. Two sub-variants are tested: **MemOS-V**, which uses a vector database for memory storage, and **MemOS-G**, which employs a graph database. More information about methods in Appendix E. All experiments are conducted using `Qwen3-8B` on both EgoMemBench and LifeMemBench. To investigate how model size impacts memory system performance, we further perform comparative experiments with `Qwen3-32B` and `Qwen3-Plus`[2]. For any embedding-related requirements, we use `Qwen3-embedding-8B`. Experiments are conducted under both offline and online modes.

### 4.2 RESULTS AND ANALYSIS

**Comparison across methods.** Table 3 presents the experimental results of various memory systems using `Qwen3-8B` as backbone across both benchmarks. The results indicate that MemOS consistently outperforms other methods, with its vector database variant (MemOS-V) achieving better

---

[2]`Qwen3-Plus` corresponds to `qwen3-plus-latest`

Table 3: Main results of memory systems' performance on two evaluation datasets. The method with the best overall performance are **bold**, the second are underlined. Within the same method, the question type with the lowest score is marked as ​xx.xx​ , the second is marked as ​xx.xx​

(a) EgoMemBench question type performances on Qwen3-8B

| Method | QT1 | | QT2 | | QT3 | | QT4 | | overall | |
|--------|-----|-----|-----|-----|-----|-----|-----|-----|---------|---------|
| | online | offline | online | offline | online | offline | online | offline | online | offline |
| RAG | 53.59 | 49.49 | 67.42 | 52.69 | 56.46 | 52.83 | 35.52 | 32.81 | 53.27 | 49.49 |
| A-Mem | 60.92 | 56.73 | 66.36 | 47.42 | 53.49 | 56.92 | 39.21 | 43.44 | 55.03 | 51.13 |
| Mem0 | 49.78 | 40.82 | 51.91 | 52.81 | 40.82 | 48.88 | 30.54 | 21.49 | 43.29 | 41.84 |
| MemOS-V | 72.74 | 70.57 | 82.43 | 80.03 | 69.44 | 68.77 | 45.98 | 45.48 | **67.59** | **66.16** |
| MemOS-G | 71.89 | 57.64 | 81.42 | 61.24 | 64.52 | 55.45 | 42.89 | 32.79 | 65.12 | 51.75 |

(b) LifeMemBench question type performances on Qwen3-8B

| Method | QT1 | | QT2 | | QT3 | | QT4 | | overall | |
|--------|-----|-----|-----|-----|-----|-----|-----|-----|---------|---------|
| | online | offline | online | offline | online | offline | online | offline | online | offline |
| RAG | 59.45 | 56.59 | 80.77 | 77.27 | 53.44 | 53.86 | 40.96 | 38.18 | 58.66 | 56.47 |
| A-Mem | 57.95 | 65.45 | 65.00 | 66.80 | 65.00 | 66.13 | 32.95 | 32.95 | 55.23 | 57.84 |
| Mem0 | 61.14 | 59.31 | 46.35 | 45.90 | 37.81 | 36.13 | 26.14 | 24.77 | 43.86 | 41.53 |
| MemOS-V | 75.00 | 70.68 | 82.50 | 81.36 | 68.41 | 66.59 | 48.18 | 48.86 | 68.52 | **66.87** |
| MemOS-G | 73.18 | 67.50 | 84.09 | 67.73 | 72.27 | 63.64 | 46.59 | 42.27 | **69.03** | 63.18 |

performance than the graph database variant (MemOS-G). Notably, neither A-Mem nor Mem0 outperform the simple RAG baseline. Diving into their methodology, both A-Mem and Mem0 rely on LLMs to summarize lifelogs and extract what the models deem "useful" memories, which inadvertently discards critical information such as timestamps, event details, and key evidence necessary for multi-hop reasoning. In contrast, RAG directly stores raw lifelog text chunks, preserving maximal original information. MemOS, however, achieves the strongest overall performance by explicitly incorporating detailed information through prompting, segmenting individual events into multiple memory units, and organizing them via a memory operating system.

**Performances of different question type.** Table 3 presents the performance of different memory systems across the four question types in LifeMemBench. Among these types, multi-hop reasoning and temporal information QA pose the greatest challenges. Temporal information QA is uniquely demanding, as queries are tightly tied to the timestamps of event logs. While most systems can readily locate the relevant event, they fail to answer correctly if time-related information is not preserved. The results show that many systems performs poor on this task—particularly Mem0 and A-Mem, whose accuracy is only slightly higher than random guessing. In contrast, MemOS and RAG achieve stronger performance, as both are designed to store and retrieve memories with explicit timestamp annotations. Multi-hop event reasoning is another challenging task, not only in LifeMemBench but also in other memory benchmarks. A key distinction from temporal information QA emerges here: A-Mem outperforms RAG on this task. This is likely because A-Mem organizes memories into more logically structured chunks, whereas RAG simply stores raw text without such structural optimization.

**Online vs. Offline evaluation.** Online evaluation more closely mirrors real-world deployment conditions. Across methods, we observe that online performance is generally higher than offline, primarily because the online setting maintains a smaller memory pool at each step, thereby reducing interference from irrelevant memories during retrieval. This suggests that current memory systems may achieve better practical performance when deployed in real-world scenarios. Furthermore, the online setting introduces a dynamically increasing level of difficulty over time, as the continual accumulation of memories poses greater retrieval challenges compared to the static nature of offline evaluation. This property makes online evaluation particularly valuable for assessing long-term memory retention and adaptability.

**Model capabilities comparison.** Under a certain capability threshold, stronger LLMs as the backbone of the memory system would achieve better performance than weaker ones, for example, `Qwen3-32B`/`Qwen3-Plus` compared to `Qwen3-8B`. However, there is no significant gap between `Qwen3-32B` and `Qwen3-Plus`. This indicates that a more powerful LLM does not necessarily result in a more powerful memory system.

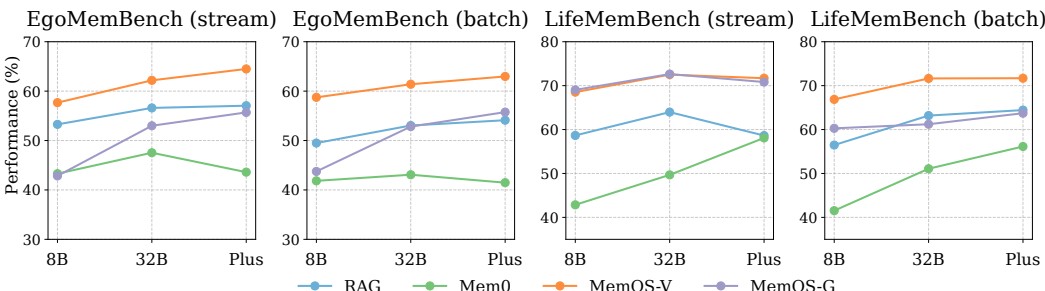

Figure 5: Comparison on the performances of memory systems using different backbone LLMs.

### 4.3 OTHER EXPLORATION

**Impact of Segmentation Granularity and Strategy on Performance.** We investigate how different segmentation granularities and event-based segmentation affect performance. In LifeMemBench, an event on average spans 13.7 dialogue turns, so we choose 8, 14 and 32 turns as the segmentation granularities, as well as event-level and day-level. Table 4 shows that event-level segmentation performs the best, while other granularities did not perform any regular pattern. Therefore, it would be better to manually train a model for event segmentation in memory system.

**Impact of Top-$k$ on Performance.** Figure 6 shows how the number of retrieved memories (top-$k$) affects the system's performance in MemOS evaluated on LifeMemBench using `Qwen3-8B`. It is obvious that the accuracy improves steadily for both MemOS-V and MemOS-G as $k$ increases, under both online and offline evaluation protocols. This demonstrates that increasing $k$ is a simple yet effective strategy to boost memory-augmented system performance. However, in practice, LLMs are often constrained by the inference latency of long-context. Therefore, we suggest setting a larger $k$ as if the latency is bearable.

Table 4: Comparison across different segmentation granularities. The best scores are **bold**, the second are underlined.

| Method | Eval Setup | 8-turn | 14-turn | 32-turn | event | day |
|--------|-----------|--------|---------|---------|-------|-----|
| **MemOS-V** | online | **69.77** | 66.92 | 66.16 | 67.79 | 65.40 |
| | offline | 65.93 | 65.81 | 64.76 | **68.38** | 64.59 |
| **MemOS-G** | online | 65.58 | 63.60 | 63.72 | 66.80 | **68.32** |
| | offline | 46.19 | 54.22 | 47.06 | **56.61** | 46.13 |

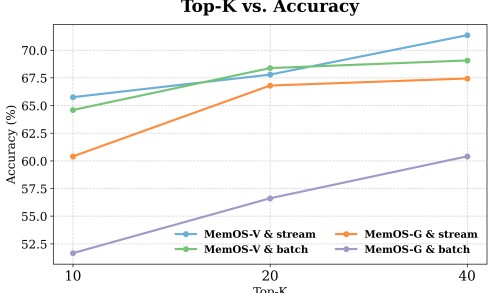

Figure 6: Comparison on accuracies of different retrieval Top-k in MemOS.

### 5 CONCLUSION

This paper addresses a critical gap in memory system evaluation by focusing on continuous dialogue lifelog scenarios—a highly promising real-world application where memory systems must handle unbroken, timestamped, and context-rich daily interactions (e.g., from always-on wearable microphones). To fill this void, we construct two complementary benchmarks with hierarchical life simulation framework, including **EgoMemBench** and **LifeMemBench**, supported by extensive experiments and analysis. Moreover, we design four targeted question types to comprehensively test memory system capabilities in lifelog scenarios. Among these, Temporal Information QA emerges as uniquely critical to lifelog scenarios—it requires systems to preserve and retrieve exact timestamps. In evaluation, we are the first to propose a online mode to align with the real-world streaming dialogue. In experiments, MemOS consistently outperforms all other three methods, while A-Mem and Mem0 do not even outperform a simple RAG baseline. This highlights a critical pitfall: aggressive summarization discards critical information that is indispensable for lifelog QA, whereas RAG's raw text storage and MemOS's structured preservation better retain this context. In summary, this work defines the continuous dialogue lifelog scenario as a critical testbed for memory systems, and offers actionable insights for designing memory systems that excel in real-world lifelog contexts. As wearable devices and terminal AI assistants increasingly adopt always-on sensing, our benchmarks and findings lay the groundwork for developing memory systems that can reliably support long-term, context-aware user interactions—positioning lifelog-aware memory as a key feature of next-generation AI.

ETHICS STATEMENT

This work focuses on routing strategies and evaluation frameworks for collaborative LLM systems. We do not involve sensitive personal data, human subjects, or potentially harmful content. Our methods aim to improve efficiency and robustness without introducing new ethical risks.

REPRODUCIBILITY STATEMENT

We provide detailed descriptions of our framework, metrics, and experimental setup in the main text and appendix. All datasets used are publicly available, and we will release code, training scripts, and evaluation pipelines to ensure full reproducibility.

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

## A    LLM USAGE

In the preparation of this paper, large language models (LLMs) were used solely as auxiliary tools. Specifically, we employed LLMs for grammar correction and text polishing, as well as to support dataset generation and assist in the manual review of data quality.

## B    DETAILS OF HUMAN-IN-THE-LOOP REVIEW

**Overall Procedure and LLM-Assisted Inspection.** The overall procedure begins with *monthly-level summary*, where annotators perform comprehensive reading, inspection, and revision. This is followed by *weekly-level summary*, which involves several checks: **(2.1) Consistency between parent and child summaries** verifies that weekly content does not contradict monthly content (e.g., ensuring events are not mistakenly placed before a meeting); **(2.2) Factual correctness** checks for obvious factual errors (e.g., accurately reflecting the initials on a ring); **(2.3) Repetition checking** uses an LLM to extract event descriptions, retrieves the five most similar events via similarity search, and inspects them to prevent unreasonable repetition (e.g., the protagonist reading the same book chapter and having identical reflections in different months); and **(2.4) Random sampling**, where 20 revised summaries are randomly selected for additional verification. The *day-level and event-level summaries* follow the same checking procedure as the weekly-level summaries. Given the substantial volume of content at the weekly, day, and event levels, we employ an LLM (specifically, Qwen3-235B-Instruct) to conduct a first-pass review for detecting potentially problematic segments, after which human annotators perform full manual inspection and correction.

**Issue Detection and Final Quality.** The proportions of issues detected by the LLM during the initial review are as follows: for parent-child consistency, weekly (11%), day (14%), and event (16%); for factual correctness, weekly (13%), day (17%), and event (19%); and for repetition checking, weekly (7%), day (10%), and event (12%). Following revisions based on these checks and subsequent human review, the final quality is confirmed through random sampling, which yields a pass rate of 100% for the weekly, day, and event-level summaries. All review work is conducted by several data annotators within our team.

## C    BENCHMARK DATA SAMPLES

### C.1    EXAMPLES OF DATASET

In this section, we provide illustrative snippets from the LifeMem Dataset. The EgoMem Dataset adopts the same formatting and structural schema.

Table 5: Jeremy and Jane at Home Organizing Old Items (2024-01-01)

| Time | Speaker | Utterance |
|------|---------|-----------|
| [08:10:15] | **Jane** | All done eating. I'll go clear the bowls first, and then shall we tidy up the cabinet in the living room? |
| [08:10:22] | **Jeremy** | Okay, I'll help you clear up. No point letting them pile up. |
| [08:10:30] | **Jane** | Yeah, and give the tablecloth a good shake while you're at it, there are some breadcrumbs. |
| [08:10:38] | **Jeremy** | Alright, you go change into some clothes you don't mind getting dirty. I'll be over as soon as I finish here. |
| [08:11:05] | **Jane** | Hey, the bottom drawer of the cabinet is stuck. Can you give it a pull? |
| [08:11:10] | **Jeremy** | Let me see... Push it in a bit, then give it a sharp tug – There, it's open. |
| [08:11:18] | **Jane** | Wow, how did this box get so dusty? I think it's the old photo albums, right? |
| [08:11:24] | **Jeremy** | Should be. That was before we switched to a digital camera, all these were developed from film. |
| [08:11:30] | **Jane** | This one... was our first trip to Hangzhou? You were wearing that blue checkered shirt that year. |

Continued on next page...

**Table 5 – continued from previous page**

| Time | Speaker | Utterance |
|---|---|---|
| [08:11:36] | **Jeremy** | Haha, yes, taken at the entrance of Lingyin Temple. You insisted that monk was peeking at us while we took the picture. |
| [08:11:42] | **Jane** | He was looking! And you started laughing, the photo turned out all blurry. |
| [08:11:50] | **Jeremy** | Check the back, I think there are some from that Yunnan trip too? |
| [08:12:00] | **Jane** | Yes, here they are. At the gate of Dali Old Town, you had to wear your sunglasses crooked, trying to look all artsy. |
| [08:12:06] | **Jeremy** | That was called 'creating a vibe'. Look how happy you're laughing in this one. |
| [08:12:12] | **Jane** | Hmm... My hair wasn't gray back then. |
| [08:12:20] | **Jeremy** | It wasn't that long ago, was it? Seven or eight years? |
| [08:12:25] | **Jane** | Almost. Time really flies. Oh, how did this USB drive box get here too? |
| [08:12:32] | **Jeremy** | Used that for storing photos ages ago. I think it's labeled "2016 Family Photos". |
| [08:12:38] | **Jane** | Can we still read it? Should we find a computer and try? |
| [08:12:42] | **Jeremy** | I'll try it later on my study computer. The port should still be compatible. |
| [08:12:50] | **Jane** | No rush, let's sort these albums first. The old ones go on this side, the newer ones over here. |
| [08:13:00] | **Jeremy** | This yellow-edged one was from your mom, right? She said we should pick only the best ones to develop and keep. |
| [08:13:06] | **Jane** | Yes, she kept saying back then that when we got old, we could look through them together. |
| [08:13:12] | **Jeremy** | She was right. Isn't it nice looking through them now? |
| [08:13:20] | **Jane** | Mmm... This box also has a group photo from Weizhou's wedding. |
| [08:13:26] | **Jeremy** | Oh, look at him in the suit with a bow tie, like he stepped right out of a period drama set in the Republic of China era. |
| [08:13:32] | **Jane** | You're one to talk! Your tie was crooked, and he had to retie it for you. |
| [08:13:38] | **Jeremy** | Haha, you remember everything. We gotta keep this photo to tease him with next time we see him. |
| [08:13:45] | **Jane** | Don't overdo it. He's "Boss Zhang" now, you know. |
| [08:13:50] | **Jeremy** | To me, he'll always be that goofball who fell into the flowerbed playing basketball. |
| [08:14:00] | **Jane** | Oh, this one is of your dad fixing his bike in the yard... |
| [08:14:06] | **Jeremy** | Yeah, that old Phoenix brand bike. The chain kept falling off, he'd spend the whole afternoon fixing it. |
| [08:14:12] | **Jane** | He was so handy. All your repair skills, you learned from him. |
| [08:14:18] | **Jeremy** | Yeah... This is a really good photo. The light on his face, so peaceful. |
| [08:14:25] | **Jane** | Let's not throw these old things away. Let's find a box and store them properly. |
| [08:14:30] | **Jeremy** | Okay, I'll go get a storage bin from the storage room later. |

Table 6: Jeremy with Family Watching Spring Festival Gala (2024-02-10)

| Time | Speaker | Utterance |
|---|---|---|
| [16:00:12] | **Jeremy** | Mom, Jane, the Spring Festival Gala replay has started. The tea is freshly brewed, have it while it's hot. |
| [16:00:18] | **Mother** | Oh, this tea aroma is so comforting. Hangzhou Longjing really is something else. |
| [16:00:25] | **Jane** | Mmm, so light and refreshing. One sip and I feel completely relaxed. |
| [16:03:40] | **Mother** | These hosts look the same as always, wearing red dresses every year, smiling like flowers. |
| [16:05:10] | **Jeremy** | Mom, don't just look at what they're wearing. There's a cross-talk performance later, you love those. |
| [16:08:33] | **Jane** | I recognize this skit actor. He was hilarious last time playing that delivery guy. |
| [16:12:15] | **Mother** | Oh my, this kid acts so well, the way he talks is exactly like Auntie Wang next door back in my hometown. |

**Table 6 – continued from previous page**

| Time | Speaker | Utterance |
| --- | --- | --- |
| [16:18:44] | **Jeremy** | Here, Mom, let me top up your tea. Careful, don't spill. |
| [16:19:01] | **Jane** | Did Dad used to love watching the Gala too? I remember you saying he always liked memorizing the punchlines. |
| [16:19:10] | **Mother** | Oh yes, your father-in-law would even take notes in a little notebook, saying he'd tell the students when school started. |
| [16:25:20] | **Jeremy** | This cross-talk is okay, but not as good as last year's. |
| [16:27:05] | **Jane** | Don't be so picky. Just being able to sit and watch it together as a family is nice enough. |
| [16:30:18] | **Mother** | Oh, speaking of cross-talk, it just reminded me—when Mingyuan was little, he went to pick bayberries on the hill behind the village. He fell out of the tree but insisted he didn't! |
| [16:30:30] | **Jeremy** | Mom, not this story again... |
| [16:30:33] | **Jane** | Huh? Tell me, tell me! I haven't heard this one! |
| [16:30:38] | **Mother** | That day, he insisted the sweetest bayberries were on the highest branch. Well, his hand slipped, and he landed right on his backside. Came back still stubbornly saying "I didn't cry," but his face was all swollen. Saying that with one side of his face puffed up, he looked like a little steamed bun. |
| [16:31:05] | **Jane** | Huh? Stung by a bee? Did you just say a bee? |
| [16:31:08] | **Mother** | Oh yes, right, it was a bee! I got mixed up—that was another time! Picking wild strawberries, there was a beehive in the grass, "buzz" and it stung him right on the face! |
| [16:31:18] | **Jeremy** | I really didn't cry, it's just... the tears came out on their own. |
| [16:31:22] | **Jane** | Hahaha, stop it! "Tears came out on their own"? What's that if not crying? |
| [16:31:27] | **Mother** | Exactly! He was so swollen even your dad couldn't recognize him, still insisting "I didn't cry." I put a cold towel on his face, and he's sniffling, saying "It's just a little itchy." |
| [16:31:40] | **Jane** | That's adorable! I have to write this down—(sound of typing on phone) Title it "Future Parenting Material". |
| [16:31:48] | **Jeremy** | Hey, don't write that down. What kind of positive example is that... |
| [16:31:52] | **Mother** | Why not? Stubborn kid, full of spirit! Kids these days don't have that kind of grit anymore. |
| [16:32:10] | **Jane** | When we... if we have kids in the future, I'll tell them this story. I'll add a subtitle: "On the Art of Graceful Stubbornness". |
| [16:32:18] | **Jeremy** | Don't you two gang up on me... |
| [16:32:25] | **Mother** | This isn't ganging up, it's family memories! Come on, Mingyuan, pour some more tea, let's keep watching. |
| [16:35:40] | **Jane** | This dance is so beautiful, the backdrop looks like an ink wash painting. |
| [16:36:15] | **Mother** | Yes, the costumes are lovely too, the colors are elegant, not too flashy. |
| [16:40:30] | **Jeremy** | The special effects here are used quite cleverly, they sync up well with the performers' movements. |
| [16:42:10] | **Jane** | See, isn't this what you called "cross-boundary integration"? |
| [16:42:15] | **Jeremy** | Heh, I guessed the start, but I didn't expect the effects to be this smooth. |
| [17:00:20] | **Mother** | This song is sung so beautifully, warms your heart listening to it. |
| [17:05:35] | **Jane** | This skit is starting to get interesting. This dad acts exactly like the department head at my clinic. |
| [17:10:12] | **Jeremy** | Shh—the accompaniment is coming up, I really like this melody. |
| [17:30:45] | **Mother** | Oh my, it's almost six o'clock. Shouldn't we start preparing dinner? |
| [17:31:00] | **Jeremy** | No rush. I've got some chicken soup with Chinese yam simmering, just need to heat it up, and there are dumplings too. |
| [17:31:10] | **Jane** | I'll set the table and pan-fry the leek dumplings we made yesterday. |
| [17:31:18] | **Mother** | Good, I'll help you with the tea. Time just flies when you're drinking this tea. |
| [19:02:10] | **Jane** | The song and dance numbers on the Gala are one after another, it's making me sleepy. |
| [19:02:25] | **Mother** | Yes, when I was young I could stay up until midnight, but now I feel like closing my eyes past nine. |

**Table 6 – continued from previous page**

| Time | Speaker | Utterance |
|------|---------|-----------|
| [19:03:05] | **Jeremy** | How about we take a break? We can get up again closer to midnight? |
| [19:03:12] | **Jane** | Okay, I'll go charge my phone first, and I need to organize my notes. |
| [19:03:20] | **Mother** | I'll just stay put here. You two go ahead, I'll just listen to the Gala. |

Table 7: Jeremy in Emergency Project Post-Mortem Meeting (2024-06-03)

| Time | Speaker | Utterance |
|------|---------|-----------|
| [10:15:00] | **Jeremy** | Is everyone here? Let's get started. As you all saw, last night's incident had a significant impact. We need to quickly piece together the timeline and identify the root causes. |
| [10:15:45] | **Mike** | Yes, we came straight from the morning stand-up. Wei and the Ops reps are here too. |
| [10:15:55] | **Jeremy** | Good. Let me briefly recap the timeline. Last night at 21:47, our monitoring platform started receiving a flood of 503 errors, concentrated on the user login and permission verification APIs. Frontend service response times spiked from an average of 80 milliseconds to over two seconds, lasting roughly twenty minutes. |
| [10:17:10] | **Alex** | On the backend side, we didn't receive alerts until 21:49, two minutes after the problem started. Furthermore, the initial alerts were scattered; no one realized it was a systemic issue initially. |
| [10:17:45] | **Wei** | The test environment monitoring didn't trigger because we hadn't simulated failure states for that authentication component. It appears a vulnerability in the third-party SDK was triggered by a scanning tool, causing it to crash outright, which then cascaded to our authorization service. |
| [10:18:35] | **Other** | Correct. Checking the logs confirms it's the CVE-2024-3187 mentioned in their urgent patch bulletin – a high-severity privilege escalation vulnerability. When their service restarted, our persistent connections were all severed, and we lacked reconnection safeguards. |
| [10:19:40] | **Jeremy** | So, fundamentally, it wasn't our code at fault. But the core issue is that our monitoring didn't flag the anomaly immediately. From 21:47 to 21:58 – a full 11 minutes – there was no clear, high-severity "service meltdown" alert. |
| [10:20:35] | **Mike** | That's unacceptable. Users couldn't access the app, and we were in the dark? |
| [10:20:50] | **Jeremy** | Exactly. Reviewing the Grafana dashboards, while we had heartbeat metrics, we lacked aggregated alerting for them. Also, the alert rules are too fragmented; a sea of red dots ended up masking the critical issue. |
| [10:21:45] | **Wei** | I checked the logs last night. The first call was to Alex at 21:55, reporting login timeouts. That's when we first suspected a common problem, but the command chain was unclear – no one took clear ownership of the emergency response. |
| [10:22:30] | **Alex** | I was initially checking logs, thought it might be a database issue, and even had the DBA team investigate. It took time to realize the upstream auth service was the root cause. |
| [10:23:15] | **Ops Rep** | We were also reactive. By the time we noticed the abnormal traffic drop and intervened, the golden window for mitigation had passed. |
| [10:24:00] | **Jeremy** | Therefore, while the trigger was a third-party component failure, this incident exposed our own weaknesses: insufficient monitoring sensitivity and a lack of a formalized emergency response process. |
| [10:24:50] | **Mike** | Agreed. The responsibility for the cause isn't ours, but our response was too slow. This has to change. |
| [10:25:15] | **Jeremy** | I propose we focus on two key areas moving forward. First, integrate health checks for external dependencies into our core monitoring. Heartbeat, version status, abnormal reconnection states – all need real-time, prominent alerting. |
| [10:26:20] | **Wei** | We can integrate that with our existing component health dashboard. Wasn't that already in progress? |

Continued on next page...

**Table 7 – continued from previous page**

| Time | Speaker | Utterance |
|------|---------|-----------|
| [10:26:40] | **Jeremy** | Yes, this fits perfectly. Second, I've been thinking since last night: we need to prioritize implementing a robust canary release and automated rollback mechanism. If we could have automatically detected the spike in abnormal call rates and rolled back to the previous stable version, we could have halved the outage duration. |
| [10:27:55] | **Alex** | Automated rollback? Isn't that a bit aggressive? What about false positives? |
| [10:28:20] | **Jeremy** | Not a full, automatic rollback for all traffic. We can start with a canary release for a small percentage of users, say 1%, while closely monitoring key metrics – error rate, latency, authentication failure rate. If these exceed thresholds, automatically route traffic back to the old version and trigger alerts. |
| [10:29:30] | **Ops Rep** | We support this approach. We can configure the traffic switching using K8s; we've tested similar setups in our test environment before. |
| [10:30:10] | **Wei** | Then our release process needs updating too. The current manual tagging and manual image push is prone to missed steps. |
| [10:30:45] | **Jeremy** | Exactly. I want to implement a pre-release checklist, similar to the one we drafted earlier. Items like dependency scans, permission verification, rollback plan confirmation – all must be checked off before deployment. |
| [10:31:40] | **Mike** | I agree with this direction. Especially regarding external dependencies, we must confirm there are no known vulnerabilities and have a degradation plan before any future deployment. |
| [10:32:25] | **Jeremy** | I'll take the lead on drafting an improvement plan covering monitoring enhancements, the release process, and the emergency response mechanism. Target is to have a first draft by the end of this week. |
| [10:33:15] | **Mike** | Okay. You coordinate. Wei, you support with testing validation. Ops team, please provide a feasibility report for the automated traffic switching. |
| [10:33:55] | **Ops Rep** | Understood. We can schedule a technical alignment meeting this afternoon. |
| [10:34:25] | **Jeremy** | Good. Additionally, I suggest we conduct a failure drill next week, simulating a third-party service outage, to test if our current response procedures can handle it. |
| [10:35:15] | **Wei** | Agreed. I'll design the scenario, maybe add some complications like alerts being incorrectly marked as low priority. |
| [10:36:00] | **Alex** | I'll prepare an emergency procedure document then, clarifying roles and responsibilities – who does what under which circumstances – to prevent the lack of leadership we saw. |
| [10:36:50] | **Jeremy** | Alright, let's proceed on that basis. We'll schedule follow-up meetings for the details. Let's wrap up this post-mortem for now? |

## C.2 EXAMPLES OF QUESTIONS

The following examples demonstrate the four distinct question types included in our benchmark, featuring the question query, associated timestamp, and candidate options.

---

**Single Event:** What was the main topic of discussion between Jeremy and Jane during the organization of old items? `[query_timestamp=2024-01-03]`

- **Options:** A. Memories of their 2018 trip to Dali and Lijiang in Yunnan; B. Preliminary planning for the Spring Festival holiday; C. Optimization solutions for household clutter management; D. Discussion on edge computing communication protocols.

---

**Event Detail:** What specific item did Jane mention when recalling the Yunnan trip? `[query_timestamp=2024-01-01]`

- **Options:** A. A tie-dyed scarf;    B. A bicycle;    C. A hat;    D. A pair of shoes.

---

**Multi Event:** During which activities did Jeremy and Jane discuss topics related to children's health? `[query_timestamp=2024-01-01]`

- **Options:** A. During breakfast and balcony reading; B. While organizing old items and watching a movie; C. During grocery shopping and dinner preparation; D. During lunch and while debugging the projector.

---

**Temporal Info:** What was the specific time when Jeremy and Jane began immersing themselves in the photos from their Yunnan trip? `[query_timestamp=2024-01-03]`

- **Options:** A. 9:00 AM;    B. 10:30 AM;    C. 11:00 AM;    D. 1:00 PM.

---

## D  PROMPTS USED TO CURATE LIFELOG

Here is the prompt we use for transforming 10-minutes summarization to lifelog:

---

**Prompt Template to Allocate**

You are required to transform the target first-person narrative into lifelog-style conversation records. **Lifelog** refers to authentic daily spoken conversations captured by portable recording devices. Your task is not storytelling but converting the given narrative into natural dialogues that sound like real speech.

# Character name
{character_name}

# Previous Narratives (context for coherence):
{previous_narratives}

# Target First-person Narrative:
{first_person_narrative}

# Time range in target narrative:
{time_range}

# Conversation Generation Requirements
**Core Conversion Principles:**
1. **Narrative-to-Lifelog Transformation**: Convert the target first-person narrative into lifelog dialogues, ensuring all important details from the narrative are preserved in the conversations.
2. **Continuity and Non-redundancy**: Previous narratives are provided to maintain timeline consistency, character relationships, and avoid repeating the same details unnecessarily.
3. **Authenticity**: The dialogues must sound natural, spontaneous, and spoken in real daily English, avoiding formal or literary expressions.

**Format Specifications:**
- Strictly use the format: [yyyy-mm-dd, HH:MM:SS] Character: Speech content

**Content Requirements:**
1. **Detail Preservation**: Every concrete detail in the target narrative (actions, observations, emotions, objects, times, etc.) must appear in the dialogues.
2. **Logical Flow**: Keep the event flow consistent with both the target narrative and previous lifelogs.
- Ensure continuity of relationships between characters.
- Keep the timeline reasonable and coherent.
3. **Boundary Control**: Do not introduce cross-day planning, greetings, farewells, or artificial summaries. End conversations naturally when the described event ends.

**Output Format:**
- Only output lifelog dialogues in English, without explanations, notes, or extra text.

# Example Format
[2025-09-17, 09:23:11] Speaker A: Actual spoken words
[2025-09-17, 09:23:15] Speaker B: Dialogue continues

Now please generate lifelog conversations according to the above requirements.

The following prompts were employed in the Top-Down Hierarchical Life Simulation Framework. Year-level summaries are progressively allocated and enriched at the month level to generate detailed monthly summaries, while the prompts for the "month-to-week" and "week-to-day" stages have been slightly adjusted.

### Prompt Template to Allocate

You are a professional lifelog analyst. Based on the provided annual experience summary, restructure and expand the content by month to generate detailed, coherent, and realistic monthly life records.

{holidays}
{important_days}

# Annual Experience Summary:
{year_summary}

# Requirements
- Each monthly record must clearly describe the time, location, people involved, process, and outcomes of events.
- While strictly reconstructing the annual experiences by month, you may expand each month's record.
- Your expansions must be realistic; ensure the content is substantial and natural, and avoid fabricated dramatic plots or supernatural elements.
- After reconstruction and expansion, each month's record must cover major events, work,

exercise, entertainment, family communication, and social activities.
- If a specific time point for an event is clearly stated in the annual summary, you must not change it; if it is not specified, assign a reasonable time.

# Output Format
Output strictly as a standard JSON array, and output only the JSON array without any explanations or comments. Each item in the JSON array should have the following structure:
[ {{ "Month": "{year} January", "Monthly Record": "..." }}, {{ "Month": "{year} February", "Monthly Record": "..." }}, ... ]

---

**Prompt Template to Enrich**

You are a professional lifelog analyst. Below are this person's monthly records for the target month and the adjacent months. Please enrich the current record for the target month to make the description more comprehensive.

{prev_months}

# Existing monthly record for {month}:
{month_data}

# Date information for {month}:
{month_dates_info}

# Requirements:
- Each monthly record must clearly describe the time, location, people involved, process, and outcomes of events.
- Unless the current month's record already contains such mentions, do not add any cross-month plans during enrichment; for example, do not schedule April activities in the March record.
- The enriched content must be realistic; ensure the content is substantial and natural. Avoid fabricated dramatic plots or supernatural elements.
- The enriched record should cover all facets of life, including but not limited to major events, work, exercise, entertainment, family communication, and social activities.
- The enriched content must cover the entire month—early, mid, and late—and distribute events as evenly as possible. If the original record provides specific dates/times, you must keep them.
- The enriched content must remain temporally consistent with the records of the previous and following months, ensuring coherence without contradictions.
- Note that workdays are typically Monday through Friday, rest days are Saturday and Sunday, and public holidays are rest days. Arrange work and life content accordingly.

# Output Format
Output strictly as standard JSON, and output only the JSON without any explanations or comments. The JSON fields are:
{ "Month": "{month}", "Monthly Record": "..." }

## D.1 QUESTION GENERATION PROMPT

The following prompt is an example prompt for daily-level question-answer pairs generation. To adapt this prompt for weekly, monthly, or other temporal granularities' generation, it only needs to adapt the description from `Daily Events` to `xx Events`.

---

**An Example Prompt Template to Generate Daily-level Questions**

# Prompt for Event Extractor Evaluation Data Generation
You need to generate evaluation data for an event extractor. The event extractor will extract useful information from users' life records and store it in a database.
Now you will be provided with a user's daily experiences, and you need to generate four questions based on the content, with four options (A, B, C, D) for each question (one correct answer and three distractor options). These questions and options will be used to evaluate the extraction performance of the event extractor.

## Daily Events (date)
{all_day_events}

## Question Requirements
- Generate 4 question-answer pairs, which should ask about the following four aspects respectively:
- The content of a specific event
- A specific detail of a specific event
- The content of multiple events
- The specific time when a specific event occurred
- Question Guidelines:
- Frame questions about events that involve interactions with others and can generate dialogue data; do not frame questions about events that cannot generate dialogue data.
- The events targeted by the questions must be unique enough and must not be daily routine events.

## Output Requirements
- You need to output a JSON list, where each JSON element contains the following fields:
- 'question': The content of the question
- 'options': A list containing four options, formatted as ["A. Option content", "B. Option content", "C. Option content", "D. Option content"]
- 'answer': The option letter of the correct answer, e.g., 'A'
- Do not output any content other than the JSON list of question-answer pairs

---

### D.2 QUESTION TYPES DISTRIBUTION

The distribution of four question types are in Figure 7. We ensure that both benchmark would have a balance question-type proportion.

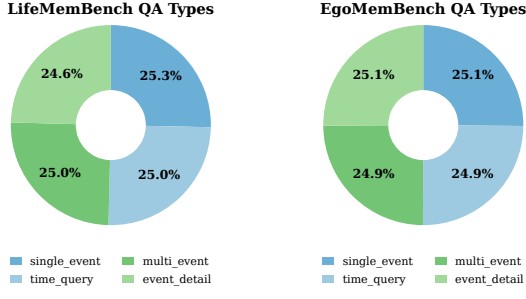

Figure 7: The Distribution of QA types.

## E  IMPLEMENTATION DETAILS ABOUT MEMORY SYSTEMS

**RAG**  The simple RAG baseline include a chat-agent and an embedding model to save and retrieve the relevant text. Therefore, no summary agent, no LLM memory manager, and no LLM retriever inside the system. It directly embed and retrieve the liflog text chunks into a vector database.

**A-Mem, Mem0 and MemOS** We follow the official code of these memory systems' on github repositories for evaluation. The prompts inside these systems are specifically refined to fit the requirement for our benchmark evaluation.

