# OpenReview forum: "Benchmarking Long-Term Memory with Continuous dialogue Lifelogs"
_ICLR.cc/2026/Conference — ICLR 2026 Conference Withdrawn Submission_

### Official Review · Reviewer_W12f · 2025-10-30

**Soundness:** 2
**Presentation:** 2
**Contribution:** 2
**Rating:** 4
**Confidence:** 4

**Summary:**

The paper proposes two benchmarks, EgoMemBench and LifeMenBench, for evaluating long-term memory systems in continuous dialogue lifelog scenarios. It also introduces a QA construction pipeline with four question types, and a novel online evaluation mode that better reflects real-world memory usage. However, there are some concerns regarding data generation and the experimental interpretation, detailed below.

**Strengths:**

1. The focus on continuous dialogue lifelogs represents a promising and underexplored scenarios. The problem is clearly defined.
2. The proposed “online” evaluation mode effectively mirrors real-world deployment, where queries are answered using current memories, while new information is continuously processed.

**Weaknesses:**

1. The experiments mainly compare the performance on four memory systems,  but do the observed results reflect genuine challenges for memory systems, or do they arise from noise or ambiguity within benchmark itself?  A comparison with existing benchmarks would strength the claims.
2. The authenticity and generalisability of the synthetic dataset are questionable, potential LLM-induced stereotypes or biases during data generation may compromise realism.
3. It would be helpful to include sample examples of the four question types for better transparency.
4. The experimental setup is limited to the Qwen3 model series. Validation on other mainstream LLMs (e.g., Claude, Llama3, and GPT-4) is missing, raising doubts about whether MemOS’s advantages is due to its architecture or its compatibility with Qwen3.
5. Table 2 shows that Qwen-3-max-preview achieves nearly 100% accuracy, suggesting that benchmark questions themselves may not be difficult. The benchmark’s ability to distinguish between advanced memory systems with different reasoning capacities is therefore uncertain.

**Questions:**

1. How are the noise and irregular data handles in the two datasets? Are the unstructured and incoherent characteristics of real-world lifelogs modeled to enhance robustness ?
2. In Table 4, the performance of different segmentation settings are very small (less than 5% for MemOS-V).  Could you explain the reason behind this?
3. Figure 5 shows that MemOS-G performs similarly under both stream and batch settings in EgoMenBench.  Why does this occur?
4. In the “online evaluation mode”, how is the query stream generated? Are these queries predefined or dynamically produced during the simulation?
5. How are queries temporally aligned in the streaming setting? What is the relationship between the query timestamp and the current memory time window?

---

### Official Review · Reviewer_FJsV · 2025-10-31

**Soundness:** 2
**Presentation:** 2
**Contribution:** 2
**Rating:** 2
**Confidence:** 4

**Summary:**

In this paper, the authors propose benchmarks for evaluating long-term memory capabilities in continuous dialogue lifelog settings. They motivate their work by pointing out the lack of benchmarks that effectively model ongoing, everyday (always-on) conversational experiences. Building on a hierarchical life simulation framework, the authors introduce two complementary benchmarks: (1) EgoMemBench, which is based on the real-world EgoLife dataset, (2) LifeMemBench, a synthetic dataset generated using LLMs. The two datasets are constructed using bottom-up and top-down approaches respectively, and include question–answer pairs designed to capture temporal changes in information over time. Using these benchmarks, the authors conduct experiments to evaluate the performance of several memory methods.

**Strengths:**

1. The paper defines a realistic continuous dialogue lifelog setting that reflects how people interact in daily life. Because the data include real timestamps and continuous updates, the benchmark helps evaluate how well a memory system can track and update information over time.
2. The benchmarks use a hierarchical life simulation framework that organizes data by time. This design allows the study of how memory systems handle information at different time scales, from short-term to long-term.
3. The authors test several well-known memory systems, such as RAG, A-Mem, Mem0, and MemOS, under both offline and online settings. These experiments show that the benchmarks are useful for comparing systems and reveal the challenges of keeping memories accurate and consistent in continuous dialogue situations.

**Weaknesses:**

1. The experiments are conducted only with the Qwen models. To strengthen the main claims, it would be better to include additional baseline models. This would help determine whether the results and analyses generalize beyond the Qwen model.
2. The paper provides no illustrative examples from the proposed datasets. Since the benchmark is a major contribution, including sample data or qualitative cases would increase clarity and help readers assess the reliability of the dataset.
3. Compared with existing long-term memory benchmarks such as LongMemEval or LoCoMo, the proposed datasets cover a shorter temporal range. While the continuous dialogue lifelog setting is meaningful, extending the duration would make the benchmark more competitive and realistic.
4. The evaluation relies solely on multiple choice questions. While this format easy to scoring, it may provide cues to the model and fail to reflect the open-ended reasoning expected from real-world memory systems. Adding an open-ended QA setting could provide a more complete and challenging evaluation.
5. The experimental section feels somewhat narrow in scope. Including ablation studies or deeper analyses, for example, testing the influence of retrieval parameters, etc., would make the results more convincing and informative.
6. Details about the annotators and the review guidelines used in the data review process (Section 3.2) are completely missing from the paper. The authors are strongly encouraged to clarify these aspects and fully address this concern during the rebuttal phase.

**Questions:**

1. Although EgoMemBench is based on the EgoLife dataset, which contains multimodal data (video), the paper uses only text. Incorporating visual information could make the benchmark more comprehensive and novel? (just question)
2. Could you provide some examples of the benchmark? It would be very helpful for me to assess the quality of the dataset.
3. As there is no human evaluation of the benchmark, the paper would better from a detailed description of the human verification pipeline. Could you provide a specific, step-by-step account of this process? Providing more details about the annotators' background, evaluation criteria, and quality control procedure would improve transparency and help assess the reliability of the human review process.

Please check the weaknesses for other questions.

---

### Official Review · Reviewer_vWTR · 2025-11-01

**Soundness:** 2
**Presentation:** 2
**Contribution:** 3
**Rating:** 4
**Confidence:** 3

**Summary:**

This paper addresses the challenge of long-term memory for AI agents in continuous dialogue lifelogs, a scenario where an always-on wearable device records daily multi-person multimodal conversations. It introduces two new benchmarks: EgoMemBench and LifeMemBench. EgoMemBench is built from a real-world egocentric dataset, covering seven days of audio and video lifelogs, totaling 300 hours. LifeMemBench is a year-long simulated lifelog generated by LLMs. From these rich lifelogs, it generates closed-ended multiple-choice QAs, and open-ended QAs. It also proposes a novel online evaluation protocol, which assesses a system’s performance incrementally as new dialogues stream in.
The benchmarks are used to evaluate four representative long-term memory systems integrated with Qwen3-8B. It also extends the model size to 32B to investigate the stronger LLMs as the backbone of the memory system.

**Strengths:**

1.	The paper identifies a critical unexplored scenario, continuous multi-person lifelogs, for long-term memory in LLMs. This setting is significant and innocuous, and the data sources from real-world for EgoMemBench, which demonstrates high quality.
2.	This paper contributes two benchmarks. EgoMemBench grounds the evaluation in real-world data, while LifeMemBench provides scale and diverse data. The data generation pipeline is well-designed. After generating conversational data, the evaluation QAs also cover various aspects. The online evaluation mode is also insightful.
3.	The evaluation on Qwen3-8B is comprehensive enough. It covers four memory systems and evaluates their performance thoroughly. It also investigates the stronger LLMs as the backbone of the memory system using the Qwen family.
4.	The paper is generally well-organized and clear. The benchmark construction is clear.

**Weaknesses:**

1.	A concern is that LifeMemBench is entirely simulated. It is acknowledged that the simulation is well-designed, and the paper has said that human evaluation is in the loop. However, a clear discussion on how humans evaluate the simulation process with a clear evaluation metric is expected.
2.	All evaluations use one LLM family (Qwen3). It is admitted that these LLMs are powerful; however,  to conduct experiments on one LLM as the main result is still not satisfactory. Moreover, recent studies have reported the performance gap between the thinking mode and non-thinking mode of the Qwen3-8B and suggest that this base model may not be stable enough. Studies on new models are necessary, and it is NOT a criticism of the base model setting as this paper is a work on benchmark; however, more experiments on the instruct models, for example, Qwen2.5-7B-instruct, may be expected.

**Questions:**

1.	If time permits, more studies on the Instruct models can help readers better understand the benchmark.
2.	LifeMemBench is OK to be entirely simulated by LLMs. However, the human-in-the-loop review should be reported clearly. Also, if using real-world data as seed data and then following your simulation process, will this better reflect the real-world lifelogs?

---

> ### Author Response · Authors · 2025-11-26
> **Response (Part 1 of 1)**
>
> We appreciate your careful review. Our primary replies are summarized below.
>
> > Q1: A concern is that LifeMemBench is entirely simulated. It is acknowledged that the simulation is well-designed, and the paper has said that human evaluation is in the loop. However, a clear discussion on how humans evaluate the simulation process with a clear evaluation metric is expected.
>
> **A1**: Thank you for your suggestion. Thank you for your suggestion. Here is the detailed workflow of our human-in-the-loop review process, which we have now included in Appendix B of the rebuttal version of our paper.
>
> **Overall Procedure and LLM-Assisted Inspection.** The overall procedure begins with *monthly-level summary*, where annotators perform comprehensive reading, inspection, and revision. This is followed by *weekly-level summary*, which involves several checks: **(2.1) Consistency between parent and child summaries** verifies that weekly content does not contradict monthly content (e.g., ensuring events are not mistakenly placed); **(2.2) Factual correctness** checks for obvious factual errors (e.g., accurately reflecting the initials on a ring); **(2.3) Repetition checking** uses an LLM to extract event descriptions, retrieves the five most similar events via similarity search, and inspects them to prevent unreasonable repetition (e.g., the protagonist reading the same book chapter and having identical reflections in different months); and **(2.4) Random sampling**, where 20 revised summaries are randomly selected for additional verification. The *day-level and event-level summaries* follow the same checking procedure as the weekly-level summaries. Given the substantial volume of content at the weekly, day, and event levels, we employ an LLM (specifically, Qwen3-235B-Instruct) to conduct a first-pass review for detecting potentially problematic segments, after which human annotators perform full manual inspection and correction.
>
> **Issue Detection and Final Quality.** The proportions of issues detected by the LLM during the initial review are as follows: for parent-child consistency, weekly (11%), day (14%), and event (16%); for factual correctness, weekly (13%), day (17%), and event (19%); and for repetition checking, weekly (7%), day (10%), and event (12%). Following revisions based on these checks and subsequent human review, the final quality is confirmed through random sampling, which yields a pass rate of 100% for the weekly, day, and event-level summaries. All review work is conducted by several data annotators within our team.
>
>
> > Q2: All evaluations use one LLM family (Qwen3). It is admitted that these LLMs are powerful; however, to conduct experiments on one LLM as the main result is still not satisfactory. Moreover, recent studies have reported the performance gap between the thinking mode and non-thinking mode of the Qwen3-8B and suggest that this base model may not be stable enough. Studies on new models are necessary, and it is NOT a criticism of the base model setting as this paper is a work on benchmark; \ac{however, more experiments on the instruct models, for example, Qwen2.5-7B-instruct, may be expected.
>
> > Q2: If time permits, more studies on the Instruct models can help readers better understand the benchmark.
>
> **A2**: Thank you for highlighting the potential stability issues with Qwen-8B. As you suggested, we repeated the same set of experiments using Qwen2.5-7B-Instruct with online mode. The results, summarized in the table below, are consistent with those reported in the paper: MemOS(-V) demonstrates strong performance. A-Mem delivers comparable performance to RAG, while Mem0 performs worse.
>
> | Models | Method | QT1 | QT2 | QT3 | QT4 | Overall |
> |:---:|:---:|:---:|:---:|:---:|:---:|:---:|
> | Qwen2.5-7B-Instruct | RAG | 58.08 | 76.88 | 45.87 | 37.98 | 54.71 |
> |  | A-Mem | 57.63 | 77.8 | 45.87 | 38.21 | 54.88 |
> |  | Mem0 | 48.74 | 55.6 | 33.71 | 32.95 | 42.76 |
> |  | MemOS | 73.1 | 78.2 | 63.23 | 46.05 | 65.06 |
>
>
> > Q3: LifeMemBench is OK to be entirely simulated by LLMs. However, \ac{the human-in-the-loop review should be reported clearly}. Also, \wa{if using real-world data as seed data} and then following your simulation process, will this better reflect the real-world lifelogs?
>
> **A3**: Thank you for your question. Regarding the details of the human-in-the-loop review, please refer to our response to Q1.
> On the matter of using real-world data as seed data, we would like to clarify that **our seed data was in fact designed with reference to real-world information**. It was manually crafted to ensure that the generated year-level summaries closely align with real-life patterns. Therefore, using an individual's actual personal experiences as seed data would not necessarily lead to better outcomes and could also raise privacy concerns.
>
> ---
>
> Thank you again for your time and effort in the review process! We remain open to any further suggestions or requests for clarification you may have.

---

### Official Review · Reviewer_QMmg · 2025-11-03

**Soundness:** 3
**Presentation:** 3
**Contribution:** 3
**Rating:** 4
**Confidence:** 3

**Summary:**

This paper addresses a critical and overlooked gap in the evaluation of memory systems for Large Language Models (LLMs): **continuous dialogue lifelogs**.
To fill this gap, the authors propose **two complementary benchmarks: EgoMemBench and LifeMemBench**.
1.  **EgoMemBench** is built "bottom-up" from a real-world, seven-day first-person video dataset (EgoLife), generating plausible dialogues from video summaries.
2.  **LifeMemBench** is built "top-down" using LLMs to simulate a year-long personal lifelog, enabling a much longer temporal span and richer simulated social interactions.

Both benchmarks utilize a **hierarchical data structure** (e.g., from seconds-to-days or years-to-days) and feature an automatic QA construction pipeline. This pipeline generates four types of multiple-choice questions (e.g., event detail, multi-hop reasoning, temporal info).

A key contribution is the introduction of an **"online" evaluation mode**, which more realistically simulates the real-world scenario of streaming data and on-the-fly querying, contrasting it with the traditional "offline" mode.

Experiments evaluate four representative memory systems (RAG, A-Mem, Mem0, MemOS):
* The **MemOS** system (particularly the vector-based MemOS-V) consistently achieves the best performance.
* A surprising and crucial finding is that **memory systems based on aggressive summarization (A-Mem and Mem0) perform worse than a simple RAG baseline**，because of summarization discarding critical details (like timestamps) necessary for answering lifelog questions.
* **Temporal Information QA (QT4)** is shown to be the most challenging task for all systems.
* **Event-level semantic segmentation** is superior to naive fixed-length chunking strategies.

Overall, this paper defines an important new direction and evaluation suite for memory systems.

**Strengths:**

1.  **Complementary Benchmark Design:** The two-benchmark approach (EgoMemBench and LifeMemBench) is very strong. EgoMemBench provides real-world grounding, while LifeMemBench provides the scale and controlled complexity (via simulation) needed to evaluate long-horizon (1-year) dependencies.
2.  **Innovative Evaluation Methodology:** The proposed "online" evaluation mode is a significant contribution. It better reflects the dynamic nature of the lifelog scenario (continuous data ingest, queries at any time) than the conventional "store-then-evaluate" offline mode.
3.  **Insightful Experimental Findings:** The results are clear and illuminating. The finding that "**summarization is harmful**" (i.e., A-Mem and Mem0 being worse than RAG) makes a powerful statement about *what* should be remembered. It highlights the critical importance of **preserving raw textual evidence** (especially timestamps and details) in the lifelog scenario.
4.  **Focus on Data Quality:** The authors detail their data curation and QA generation pipeline, including the use of LLM-assistance and human-in-the-loop review, as well as dedicated steps to filter information leakage and check for answerability. This inspires confidence in the quality of the benchmarks.

**Weaknesses:**

1.  **Synthetic Nature of LifeMemBench:** As a year-long simulated dataset, LifeMemBench is entirely LLM-generated. Despite the hierarchical design, this synthetic data may lack the organic complexity, irrationality, and "messiness" of a real human life. It risks a "model-in-the-loop" bias, where the benchmark primarily tests a model's ability to retrieve information generated by a similar model, rather than genuine human memory patterns.
2.  **Indirectness of EgoMemBench:** The dialogue data in EgoMemBench is not derived from *direct* audio transcription from EgoLife. Instead, it is *generated* from **10-minute summaries** from Ego-R1. This is a "summary-of-a-summary" approach. While the authors claim this preserves "semantic fidelity", this layer of indirection may introduce artifacts and fails to leverage the actual dialogues that may have been present in the original data.
3.  **Evaluation Format (Multiple-Choice):** The authors chose multiple-choice questions (MCQ) for "stability and usability". While practical for automated evaluation, the authors themselves concede that open-ended QA is more realistic. The MCQ format may not fully test the reasoning and generation capabilities of the systems, which might "cheat" by matching keywords from the options rather than truly "understanding" the memory.
4.  **"Online" Mode Finding:** The finding that performance is generally *higher* in the online mode is interesting, but the explanation (smaller, less noisy memory pool at each step) may also suggest that the query difficulty is not sufficiently scaled to the size of the memory bank at that point. Does the difficulty truly increase dynamically, or are the systems just getting better at retrieving (more relevant) recent memories?

**Questions:**

1.  **Re: EgoMemBench:** Could you elaborate on the decision to *generate* dialogue from Ego-R1 summaries instead of using *actual* transcribed dialogues from the EgoLife audio? Was this due to a lack of dialogue in the original audio, poor ASR (automatic speech recognition) quality, privacy concerns, or other considerations? This indirection seems to be a key design choice.
2.  **Re: LifeMemBench:** Thank you for the detailed "top-down" simulation process. How do you reason about the "ecological validity" of a year-long lifelog generated entirely by an LLM? Were measures taken to prevent the simulation from falling into an LLM's "comfort zone" (e.g., avoid generating overly repetitive or overly "logical" life events) and to ensure it is truly challenging?
3.  **Re: Evaluation Format:** Did you experiment with an open-ended "LLM-as-Gudge" version of the QA? I am curious: Mem0 and A-Mem failed on the detail-oriented MCQs, but would they perform better if the questions were higher-level, open-ended "summary" reasoning tasks (e.g., "Summarize my work priorities from last week")?

---

> ### Author Response · Authors · 2025-11-26
> **Response (Part 1 of 2)**
>
> Thank you for your careful review. Our key responses are summarized below.
>
> > Q1: Synthetic Nature of LifeMemBench: As a year-long simulated dataset, LifeMemBench is entirely LLM-generated. Despite the hierarchical design, this synthetic data may lack the organic complexity, irrationality, and "messiness" of a real human life. It risks a "model-in-the-loop" bias, where the benchmark primarily tests a model's ability to retrieve information generated by a similar model, rather than genuine human memory patterns.
>
> **A1**: Thanks for your feedback. Firstly, recent mainstream benchmarks for long-term memory, such as MemoryBank[1], LoCoMo[2], LongMemEval[3], and MemBench[4], are also "entirely LLM-generated".
> Due to the significant privacy concern, synthetic data is necessary in the early stage. Secondly, although LifeMemBench is synthetic, its seed data is carefully crafted by humans. It closely reflects the patterns of real human life, covering diverse daily scenarios, multi-person interactions, and the progression of various event lines. Moreover, the Human-in-the-loop Review stage ensures the reliability and realistic fidelity of the synthesized data. Additionally, we have included sample data from our benchmark dataset in the paper. The details can be found in **Appendix C** for your review.
>
> > Q2: Indirectness of EgoMemBench: The dialogue data in EgoMemBench is not derived from direct audio transcription from EgoLife. Instead, it is generated from 10-minute summaries from Ego-R1. This is a "summary-of-a-summary" approach. While the authors claim this preserves "semantic fidelity", this layer of indirection may introduce artifacts and fails to leverage the actual dialogues that may have been present in the original data.
>
> > Q2: Re: EgoMemBench: Could you elaborate on the decision to generate dialogue from Ego-R1 summaries instead of using actual transcribed dialogues from the EgoLife audio? Was this due to a lack of dialogue in the original audio, poor ASR (automatic speech recognition) quality, privacy concerns, or other considerations? This indirection seems to be a key design choice.
>
> **A2**: Thanks for your constructive question. We actually attempted to directly transcribe the EgoLife audio using ASR; however, the results were unsatisfactory for two key reasons.
>
> * EgoLife only includes **7 days of video data from 6 participants**, meaning the volume of dialogue captured in the final transcripts is far insufficient to support ultra-long memory.
> * Even with state-of-the-art (SOTA) ASR methods, the quality of the transcribed dialogues remains limited, which prevents us from deriving a sufficient number of meaningful questions.
>
> > Q3: "Online" Mode Finding: The finding that performance is generally higher in the online mode is interesting, but the explanation (smaller, less noisy memory pool at each step) may also suggest that the query difficulty is not sufficiently scaled to the size of the memory bank at that point. Does the difficulty truly increase dynamically, or are the systems just getting better at retrieving (more relevant) recent memories?
>
> **A3**: Compared with the Offline mode—where evaluation is performed when the memory repository is already at its maximum capacity—the memory repository in our proposed Online mode grows dynamically. As a result, the difficulty of correctly answering questions naturally increases as more memories accumulate. This allows the evaluated memory methods to experience different states of memory storage, enabling a more comprehensive assessment. Moreover, because our benchmark itself is challenging, the Online setting does not become overly easy (the accuracy achieved under the MCQ setting is only 68%).
>
> ------
>
> **Reference**：
>
> [1] [MemoryBank: Enhancing Large Language Models with Long-Term Memory](https://arxiv.org/abs/2305.10250)
>
> [2] [Evaluating Very Long-Term Conversational Memory of LLM Agents](https://arxiv.org/abs/2402.17753)
>
> [3] [LongMemEval: Benchmarking Chat Assistants on Long-Term Interactive Memory](https://arxiv.org/abs/2410.10813)
>
> [4] [MemBench: Towards More Comprehensive Evaluation on the Memory of LLM-based Agents](https://arxiv.org/abs/2506.21605)

---

> ### Author Response · Authors · 2025-11-26
> **Response (Part 2 of 2)**
>
> > Q4: Evaluation Format (Multiple-Choice): The authors chose multiple-choice questions (MCQ) for "stability and usability". While practical for automated evaluation, the authors themselves concede that open-ended QA is more realistic. The MCQ format may not fully test the reasoning and generation capabilities of the systems, which might "cheat" by matching keywords from the options rather than truly "understanding" the memory.
>
> > Q4: Re: Evaluation Format: Did you experiment with an open-ended "LLM-as-Gudge" version of the QA? I am curious: Mem0 and A-Mem failed on the detail-oriented MCQs, but would they perform better if the questions were higher-level, open-ended "summary" reasoning tasks (e.g., "Summarize my work priorities from last week")?
>
> **A4**: Thanks for your suggestions. Judging from the widely used leaderboards in the general LLMs, the advantage of the multiple-choice format (i.e., stability) far outweighs that of the open-ended format (i.e., realism). This is because the results of open-ended "LLM-as-Judge" evaluations are difficult to reproduce or align due to version changes of API-based LLMs, which undermines the core usability of leaderboards.
> Additionally, the open-ended evaluation format incurs significantly higher costs in terms of both time and money.
>
> However, we acknowledge that the open-ended format is complementary due to its realism. Therefore we further conduct the experiment of open-ened format. Specifically, we evaluate gpt-4o-mini and Qwen3-8B on LifeMemBench, using Qwen-32B as the judge model (Online Mode). The experimental results are shown in the following table. While MemOS(-V) loses its performance edge in this scenario compared to the MCQ setting, most mainstream memory methods still generally underperform compared to RAG. Notably, Mem0's accuracy drops to a considerably low level, creating a pronounced performance gap with other approaches. This decline suggests that extracting facts in a fragmented manner leads to significant information loss.
>
> | Models | Method | QT1 | QT2 | QT3 | QT4 | Overall |
> |:---:|:---:|:---:|:---:|:---:|:---:|:---:|
> | gpt-4o-mini | RAG | 36.44 | 70.25 | 19.95 | 22.65 | 37.33 |
> |  | A-Mem | 33.25 | 71.39 | 19.26 | 21.28 | 36.30 |
> |  | Mem0 | 7.28 | 20.59 | 5.27 | 8.46 | 10.4 |
> |  | MemOS | 34.25 | 60.43 | 17.67 | 13.02 | 31.22 |
> | Qwen3-8B | RAG | 31.43 | 68.19 | 25.91 | 20.82 | 36.59 |
> |  | A-Mem | 30.06 | 66.13 | 19.49 | 20.59 | 34.07 |
> |  | Mem0 | 7.97 | 35.69 | 7.79 | 10.06 | 15.38 |
> |  | MemOS | 39.77 | 62.09 | 23.95 | 19.07 | 36.11 |
>
> > Q5: Re: LifeMemBench: Thank you for the detailed "top-down" simulation process. How do you reason about the "ecological validity" of a year-long lifelog generated entirely by an LLM? Were measures taken to prevent the simulation from falling into an LLM's "comfort zone" (e.g., avoid generating overly repetitive or overly "logical" life events) and to ensure it is truly challenging?
>
> **A5**: We understand the reason for your concern. Indeed, we do encounter situations where, when the protagonist needs to perform the same type of activity at different times and we ask an LLM to fill in the details, the model may still exhibit a tendency to generate identical details even when the inputs are not the same. For example, on days 1/10 and 1/20 the protagonist may both be scheduled to watch a movie, and the LLM might decide that the protagonist watches the same movie, such as Interstellar. However, watching the same movie twice within ten days may not be very realistic.
>
> **In practice, such cases do occur but are relatively infrequent**, because our top-down hierarchical lifelog generation framework substantially alleviates the issue of repetitive content: the inputs provided to the LLM for different events differ significantly. We typically resolve the remaining issues during the human-in-the-loop review stage, where one of the core tasks is to examine whether any unreasonable repetitive details appear.
>
> ---
>
> Thank you again for your time and effort in the review process! We remain open to any further suggestions or requests for clarification you may have.

---

### Note · Authors · 2025-12-22

I have read and agree with the venue's withdrawal policy on behalf of myself and my co-authors.